# Molecular Cytological Analysis and Specific Marker Development in Wheat-*Psathyrostachys huashanica* Keng 3Ns Additional Line with Elongated Glume

**DOI:** 10.3390/ijms24076726

**Published:** 2023-04-04

**Authors:** Jingyu Pang, Chenxi Huang, Yuesheng Wang, Xinyu Wen, Pingchuan Deng, Tingdong Li, Changyou Wang, Xinlun Liu, Chunhuan Chen, Jixin Zhao, Wanquan Ji

**Affiliations:** 1State Key Laboratory of Crop Stress Biology for Arid Areas and College of Agronomy, Northwest A&F University, Xianyang 712100, China; 2Shaanxi Research Station of Crop Gene Resources and Germplasm Enhancement, Ministry of Agriculture, Xianyang 712100, China

**Keywords:** long glume, *Psathyrostachys huashanica* Keng, molecular cytogenetics, RNA-seq, marker development

## Abstract

*Psathyrostachys huashanica* Keng (*2n* = 2x = 14, NsNs) is an excellent gene resource for wheat breeding, which is characterized by early maturity, low plant height, and disease resistance. The wheat-*P. huashanica* derivatives were created by the elite genes of *P. huashanica* and permeate into common wheat through hybridization. Among them, a long-glume material 20JH1155 was identified, with larger grains and longer spike than its parents. In the present study, the methods of cytological observation, GISH, and sequential FISH analysis showed that 20JH1155 contained 21 pairs of wheat chromosomes and a pair of *P. huashanica*. There were some differences in 5A and 7B chromosomes between 20JH1155 and parental wheat 7182. Molecular marker, FISH, and sequence cloning indicated 20JH1155 alien chromosomes were 3Ns of *P. huashanica*. In addition, differentially expressed genes during immature spikelet development of 20JH1155 and 7182 and predicted transcription factors were obtained by transcriptome sequencing. Moreover, a total of 7 makers derived from Ph#3Ns were developed from transcriptome data. Taken together, the wheat-*P. huashanica* derived line 20JH1155 provides a new horizon on distant hybridization of wheat and accelerates the utilization of genes of *P. huashanica*.

## 1. Introduction

Wheat (*Triticum aestivum* L., *2n* = 6x = 42, AABBDD) is one of the most significant crops, which contributes about 20% of the total dietary calories and proteins worldwide [1]. The way of distant hybridization is terrific for common wheat to obtain some distinct characteristics [2,3]. Wheat would have three gene pools as the gene pool concept [4]. The tertiary gene pool has no genome of common wheat, but they comparably contribute to wheat breeding, including Rye (*2n* = 2x = 14, RR), *Leymus mollis* (*2n* = 4x = 28, NsNsXmXm), *Thinopyrum intermedium* (*2n* = 6x = 42, JJJ^S^J^S^StSt) and *Psathyrostachys huashanica* keng (*2n* = 2x = 14, NsNs). The superior traits of wheat relatives (tertiary gene sources) were introduced into wheat breeding by distant hybridization and chromosome engineering, which provided candidate genes for wheat genetic improvement and enriched the genetic diversity of wheat [5].

*Psathyrostachys huashanica* Keng (*2n* = 2x = 14, NsNs) is an endangered wheat-related species distributed in the Huashan region of Shaanxi province, China [6]. *P. huashanica,* with abundant superior germplasm of stress-bearing, disease resistance, and early-maturing, belongs to the tertiary gene pool of wheat. Researchers have developed a host of wheat-*P. huashanica* derived lines as intermediate materials for wheat improvement, including additional lines, substitution lines, and introgression lines. So far, some good agronomic traits from *P. huashanic* have been introduced into common wheat, like the disease resistance of powdery mildew, wheat take-all [7,8], leaf rust [9], and stripe rust [10].

The glume development stage plays an important role in the growth stage of wheat. With floral organs growing, every little spike has a differentiation of two glumes and some florets [11]. The glume is protective like a shelter, and there are florets surrounded by lemma and palea inside [12,13]. Previous studies revealed that glume phenotype was correlated with an increase in spike length, grain length, and thousand-grain weight (TGW) while with a decrease in fertility, grain number, and awn length [14,15,16]. The most well-known kinds of wheat with elongated glume are *T. polonicum* and *T. petropavlovskyi* (also called “Daosuimai”) [17,18]. Researchers described that the gene for long glume (P1) of *T. polonicum* was located on chromosome 7A and identified VEGETATIVE TO REPRODUCTIVE TRANSITION 2 (*VRT2*) encoding a MADS-box as the gene underlying the *T. polonicum* long-glume (P1) locus through map-based cloning [19,20,21]. Then the locus was further fine P1^pet^ mapped [22]. The isolation of *VRT-A2* was reported as the P1 candidate gene for encoding an SVP-clade MADS-box transcription factor [23,24]. Furthermore, 560-bp substitutes for 157-bp, leading to an ectopic expression of *VRT-A2,* facilitating the elongation of glumes and grains [23].

In our research group, a batch of wheat-*P. huashanica* derivative offspring was created by hybridizing heptaploid wheat-*P. huashanica* H8911 (2*n* = 7x = 49, AABBDDNs) with *T. durum* Trs-372. Among them, a derivative 20JH1155 with elongated glume and large kernel was screened out. Agronomic traits assessment, cytological observation, GISH and sequential FISH, molecular marker identification, sequence cloning, and transcriptome sequencing analysis were used in this study to identify chromosome composition and the possible origin of the elongated glume of 20JH1155. The aim of our study was to identify the superior traits and chromosomal composition of wheat-*P. huashanica* derived line 20JH1155, and explore the origin of the elongated glume trait so that it can provide a reference for long-glume research and wheat breeding.

## 2. Results

### 2.1. Subsection Agronomic Traits

The agronomic characteristics of 20JH1155 and its parents were measured in 2021 and 2022. The data revealed that 20JH1155 with 1.47 ± 0.7cm had longer glume than 7182 with 1.07 ± 1.0cm (Table 1). In addition to the elongated length of the glume, 20JH1155 had a longer spike length while a shorter plant height and flag leaf (Figure 1A–D, Table 1). It is worth mentioning that the grain of 20JH1155 had a big size both in length and width (Figure 1E, F, Table 1). Moreover, the disease severity assessment of fusarium head blight (FHB) at 21d after infection was estimated by infected spikelet rate (ISR) and showed that 20JH1155 (ISR = 7.92%) had a higher resistance than 7182 (ISR = 74.20%) significantly (Figure 1G).

### 2.2. Observation Cytologenetics and In Situ Hybridization of 20JH1155

A total of 107 root tip cells (RTCs) of 20JH1155 were captured in mitosis metaphase by using an Olympus BX-43 microscope. The slide observation showed that 103 (96.26%) cells contained 44 chromosomes. The proportion indicated that 20JH1155 had a chromosome number of *2n* = 44. The chromosome composition of 20JH1155 was identified by the Genomic in situ hybridization (GISH) method using the whole genome of *P. huashanica* as the probe. The result revealed that 20JH1155 had two additional alien chromosomes with green hybridization signals (Figure 2A). The signals expressed 20JH1155 had 42 chromosomes of common wheat plus 2 chromosomes of *P. huashanica*.

The probe made of oligo-primer pSc-119.2 and pTa-535 for fluorescence in situ hybridization (FISH) was able to analyze the chromosome constitution of 20JH1155. A total of 42 chromosomes can be clearly identified as common wheat chromosomes, while 2 chromosomes did not show the signals of oligo pSc-119.2 or oligo pTa-535. Sequential GISH-FISH proved that the 2 no-signal chromosomes were from *P. huashanica* (Figure 2B). The specific FISH probe HS-TZ3 and HS-TZ4 [25] made 2 chromosomes have signals of *P. huashanica,* and 42 chromosomes be no hybridization signal (Figure 2C). Since there were some differences in chromosome structure between 20JH1155 and 7182, the chromosomes of tetraploid parent *T. durum* Trs-372 were also hybridized with the FISH probe (Figure 2D). The 20JH1155 karyotype was obtained by comparing it with 7182 standard karyotypes, and it demonstrated that 20JH11155 contained all 42 chromosomes of common wheat (Figure 2E). It was found that there were two additional green spots on 5AL in 20JH1155, different from 7182 and Trs-372, which only have a pTa-535 signal. Furthermore, there were also pSc-119.2 probe signals at the end of 7BL chromosomes. There were red signals on 5AL and 7BL of 20JH1155, similar to Trs-372 (Figure 2F).

### 2.3. Molecular Marker Analysis

A total of 72 EST-STS, 131 PLUG, and 6 SCAR markers were used to analyze parental common wheat 7182, *P. huashanica*, durum wheat Trs-372, and 20JH1155. The markers of the third homoeologous showed specific bands only in *P. husanica* and 20JH1155 but not in common wheat 7182 and *T. durum* Trs-372, while other homoeologous chromosomes did not show special bands. The markers amplified specific bands included 2 PLUG markers (TNAC1267-Taq1/Hae111, TNAC1286-Taq1/Hae111), 2 SCAR markers (S3-113, S3-125) and 3 EST-STS markers (CD454742, CD454086, CD454575) (Figure 3A). The sequence alignment analysis of 20JH1155 and *P. huashanica* obtained from S3-113 illustrated the sequence of 20JH1155 was the same as *P. huashanica* and only a base different from the reported sequence R3-113 (Figure 3B, Appendix A). The result indicated that the two alien chromosomes of 20JH1155 were traced to 3Ns of *P. husanica*.

### 2.4. Identification of the Variant Section of VRT-A2 in 20JH1155

Two of the designed markers were particular primers to clone the variant section (Appendix A). Among 20JH1155 and its parental materials, only 7182 and 20JH1155 amplified bands. Sequencing results analyzed by DNAMAN showed that the sequence of 7182 could map to *TaVRT-A2* with partial differences (Figure 4A). The sequence in 20JH1155 had low or no homology with a 560-bp sequence of *TaVRT-A2,* while there was the totally same 157-bp variant sequence (Figure 4C). The deletion and insertion of the sequence enhanced the duplication of motif IME (Figure 4C, red box). Compared with 20JH1155, there was a 560-bp sequence partially homologous to CS but not the same at all with 157-bp in 7182. 20JH1155 has a 157-bp instated region while missing a 560-bp of intron-1 of *TaVRT-A2* (Figure 4B).

### 2.5. RNA Expression Level Detection

A transcriptome data of 18 samples at three time points in the booting stage of line 7182 and 20JH1155 showed that each sample had a number of genes with FPKM values more than 0.3 from 52,110 to 58,702 (Figure 5A). In the three time points, 3720 genes were expressed only in common wheat 7182, 5735 genes were expressed only in 20JH1155, and 50,822 genes were expressed in both of the two (Figure 5B)**.** There were genes expression in one stage, two stages, or three stages, and the number of genes expressed in all three periods of 20JH1155 is 2413, which were potentially exogenous genes (Figure 5C).

GO (Gene ontology) analysis was performed on 2413 potentially exogenous genes (Figure 5D). GO enrichment results showed that the stage-specific exogenous expression genes were enriched in cytochrome b6f complex assembly (GO:0010190); betaine-aldehyde dehydrogenase activity (GO:0008802); mannan endo-1, 4-beta-mannosidase activity (GO:0016985); guanyl-nucleotide exchange factor activity (GO:0005085).

The co-expression data were analyzed from genes of FPKM > 3 through the Weighted Gene Co-Expression Network (WGCNA) to explore the relevant patterns of differentially expressed genes (DEGs) between 7182 and 20JH1155. In total, 22,720 genes were analyzed by 27 module clusters (Appendix A) and 2525 genes were gathered in the brown block. The expressed 1004 DEGs in three periods of the brown module were annotated using known functional genes and transcription factor prediction (Figure 5F). A total of 112 genes were annotated (Figure 5E). Among them, 95 genes were annotated in rice functional genes, including Defective Pollen Wall 3 (*DPW3*) required for pollen wall formation, *OsBRM* controlled embryo development, *OsSAMS1* related to senescence of wheat leaves and seed size, *OsNDUFA9* been essential for embryo development and starch synthesis [26,27,28]. In addition, 37 DEGs (including 7 potentially exogenous genes) were predicted as a transcription factor (Appendix A).

### 2.6. Development and Evaluation of P. huashanica Chromosomes Molecular Makers

Comparing the data of 7182 with 20JH1155 and aligning the selected sequences to de novo assembly, a total of 1756 discrepant expressed unigenes (FPKM ≥ 1) remained. Finally, seven unique unigenes were chosen to design 12 pairs of primers. To test the accuracy of the developed markers, all of them were amplified in common wheat (7182, CS, AK58) and wheat-related species (*S. cerale*, *Ae. geniculata*, *Th. ponticum*, *Th*. *elongatum*, *Th. bessarabicum*, *P. spicata*). Among them, seven markers amplified specific bands only on *P. huashanica* and 20JH1155 with a development success rate of up to 58.3% (Figure 6; Appendix A). These markers would be more economical and efficient in detecting alien chromosomes of *P. huashanica*.

## 3. Discussion

So far, researchers have created wheat-*P. huashanica* derivatives that carry excellent characteristics, including disease resistance, early maturity, and flour quality. A wheat–*P. huashanica* derived line carried two translocations: T3DS·3DL-4NsL and T3DL-4NsS resisted to powdery mildew [29]. A wheat-*P. huashanica* 7Ns addition line conferred early maturation characteristics [30]. A 7182-1Ns additional line carried a novel high-molecular-weight glutenin subunit (HMW-GS) from *P. huashanica* [31]. In addition, the introduction of 3Ns to common wheat could improve the resistance to stripe rust and yellow rust [32]. In this study, a wheat-*P. huashanica* addition line 20JH1155 with elongated glume combined with longer spike, large grain, resistance to FHB, was screened out from the wheat-*P. huashanica* derivatives in our research group. Molecular markers proved 20JH1155 was wheat-*P. huashanica* 3Ns additional line. Because of its parental wheat varieties without long glume, it could be inferred that the elongated glume of 20JH1155 was traced to *P. huashanica.* Moreover, 20JH1155, with high thousand-kernel weight and early maturation, could contribute to chromosome engineering breeding.

In previous studies, researchers reported a 560-bp from *T. polonicum*, which is a key ON/OFF molecular switch for *VRT-A2* expression. It recruited not only transcriptional repressors but also conferred intron-mediated transcriptional enhancement [23,33]. The sequence also made effects on agronomic traits about increasing grain length and TWG besides elongated glume [14,34,35]. To compare our elongated glume materials with the sequence, we cloned and sequenced the TraesCS7A03G0411400 (IWGSC-RefSeq-V2.1). The result showed that there was a 157-bp, the same as reported in 20JH1155. Under this situation, 20JH1155 has a big grain size and high TWG, which may be connected with the variant sequence.

In recent years, methods of cytological characterization, molecular maker analysis, in situ hybridization, and SNP array have accelerated the identification of wheat-*P. huashanica* derivatives [6,36,37]. In addition, besides EST and SSR markers, the SCAR markers of 1Ns, 3Ns, and 5Ns are more specific and efficient for detecting alien chromosomes of *P. huashanica* [38,39,40]. The FISH probe karyotyping of *P. huashanica* lays a solid foundation for distinguishing 1-7Ns chromosomes [25,41]. Transcriptome sequencing technology is a quick and comprehensive method providing almost all the gene information of a species in a certain state, so it has become a trend to study genes at a transcript level [42,43]. The comparative transcriptome analysis of synthetic and common wheat in responding to salt stress indicated that salt tolerance was differentially controlled between common wheat and SH wheat [44]. The potential mechanism in storage protein trafficking within developing grains of common wheat was revealed by transcriptome analysis [45]. In the study, we used morphological observations, GISH, and sequential FISH, molecular markers to identify a wheat-*P. huashanica* 3Ns additional line. By comparing the gene expression results between 7182 and 20JH1155 at different time points of spikelet through transcriptome sequencing data, some potentially exogenous genes have been speculated.

The introduction of alien chromosomes from wheat-related species will lead to changes in the variation of wheat chromosome structure [46,47,48]. For instance, the additional 5Ns chromosomes created a wheat-*P. huashanica* T3DS-5NsL•5NsS and T5DL-3DS•3DL dual translocation line [49]. A wheat-*Th. ponticum* 1J**^s^** (1D) also made influence on 5A, 1B, 1D, and 6B chromosomes structure [50]. A wheat-*Leymus mollis* 4Ns (4D) alien disomic substitution line affected the chromosomal structure of 1A, 1D, 2B, and 5A [51]. Compared with parental common wheat, 20JH1155 has a different FISH karyotype in chromosomes 5A and 7B. These changes hinted that the alien *P. huashanica* chromosomes might influence the chromosome structure of 5A and 7B.

Developing accurate markers using transcriptome sequencing technology is more effective. Compared with traditional methods, modern sequencing technology could provide a precise and rich source for developing specific markers [52]. For example, specific markers on the *Dasypyrum villosum* chromosome were designed via genotyping-by-sequencing (GBS) [53]. A DNA marker depicted for variety discrimination specific to ‘Manten-Kirari’ based on an NGS-RNA sequence in *Fagopyrum tataricum* [54]. Single-nucleotide polymorphism markers of salinity tolerance for Tunisian durum wheat were developed using RNA sequencing [55]. Seven pairs of specific molecular markers were developed based on our transcriptome data. The markers can, directly or indirectly, expedite the process of identifying exogenous chromosomes from *P. huahsanica*.

## 4. Materials and Methods

### 4.1. Plant Materials

In this study, the materials are common wheat line 7182 (*Triticum aestivum* L., 2*n* = 6x = 42, AABBDD) combined with *Psathyrostachys huashanica* Keng (2*n* = 2x = 14, NsNs), wheat-*P. huashanica* derivative line 20JH1155 and *Triticum durum* line Trs-372 (2*n* = 4x = 28, AABB). H8911 (2*n* = 7x = 49, AABBDDNs) was obtained by hybridizing the common wheat 7182 with *P. huashanica*. Then 20JH1155 were obtained from line H8911× lineTrs-372. In addition, common wheat Chinese Spring (CS) and wheat-related species *Secale cerale* L. (*2n* = 2x = 14, RR), *Aegilops geniculata* (2*n* = 4x = 28, UUMM), *Thinopyrum ponticum* (2*n* = 10x = 70, E**^e^** E**^e^** E**^b^** E**^b^** E**^x^** E**^x^** StStStSt or JJJJJJJ**^s^** J**^s^** J**^s^** J**^s^**), *Th*. *elongatum* (2*n* = 2x = 14, E**^e^** E**^e^** or EE), *Th. bessarabicum* (2*n* = 2x = 14, E**^b^** E**^b^** or JJ) and tetraploid *Pseudoroegneria spicata* (2*n* = 4x = 28, StStStSt) were used as contrast. All the materials were preserved at the College of Agronomy, Northwest A&F University, China.

### 4.2. Assessment of Agronomic Traits and Disease Resistance

The agronomic traits and common diseases of the materials and their parents were evaluated in 2021 and 2022. The materials were planted in a randomized block with two rows of 1-m-width lines in the field of Northwest A&F University, Yangling, China. During the growth period, ten plants selected randomly would be investigated for their morphological characteristics and diseases at reasonable times. The traits for the individual plant included plant height, spike length, glume length, tiller number, spikelet number, florets number per spikelet, thousand-kernel weight, kernel length, kernel width, and awn. The kernel conditions were measured using the SC-G automated seed testing system (Wseen Detection Technology Co, Ltd., Hangzhou, China). Each sample contained around 300 kernels in the random selection and was repeated 3 times.

The FHB resistance assessment was evaluated by *F. graminearum* strain PH1. At the flowering stage, the single flower infusion method was used to infect 10 μL micro-conidial suspension (250,000 spores mL^−1^) to 10 spikes of 7182 and 20JH1155, respectively. Plastic bags were used to cover the spikes for 2 days, and the 306 disease severity would be investigated 21 days later [56]. The resistance of fusarium head blight (FHB) was estimated by the infected spikelet rate (ISR).

All the data were analyzed by SPSS Statistics software (IBM SPSS Statistics 25.0, Armonk, NY, USA).

### 4.3. Cytological Identification

To obtain the RTCs, the seeds were soaked in a petri dish with distilled water and filter paper for 24 h, then poured off excess water and put seeds in order. After that, set the dishes at 23 °C Incubator in the dark until the seed had roots at a length of 2–3 cm. Next, the root tips were cut and put in moist cuvettes in a one-to-one relationship and then treated the roots for 2 h with nitrous oxide. Later, the root tips could do the next step or be stored at −20 °C in 70% ethanol. Treating the root tips with cellulase (R-10, Yakult Japan, Tokyo, Japan) and pectinase (Y-23, Yakult Japan, Tokyo, Japan) in a 37 °C constant temperature bath for 57 min, grinding them with a pestle, slides with RTCs split phase could be scrutinized to get the chromosomes with an Olympus BX-43 microscope (Olympus Optical Co., Tokyo, Japan).

### 4.4. GISH and FISH Analysis

Choosing clearcut RTC slides and marking the split phases, the slides were treated at an ultraviolet intensity of 1250 KJ/cm^3^ by UV irradiation (Spectrolinker^TM^ XL-1500, Long Island, NY, USA) for 60 s. The GISH probe was made of purified genomic DNA of *P. huashanica* and fluorescein-12-dUPT. The FISH probes, Oligo-pSc119.2 (6-FAM-5′, green) and Oligo-pTa535 (Tamra-5′, red), were synthesized by Invitrogen Biotechnology Co., Ltd. (Shanghai, China). The steps and components referred to 328 to [57]. Finally, the slides were scanned by an Olympus BX-53 fluorescence microscope with a DP80 camera (Tokyo, Japan).

### 4.5. Molecular Marker Analysis

A total of 72 expressed sequence tag-sequence-tagged site (EST-STS), 131 PCR-based Landmark Unique Gene (PLUG), and 6 sequences characterized amplified region (SCAR) markers, which were selected by early researchers in our laboratory, were used to identify the homoeologous groups of alien chromosomes. The polymerase chain reaction (PCR) products of EST-STS maker could straight detect by 12% polyacrylamide gel electrophoresis (PAGE). The products of the PLUG maker were further cut out by processing enzymes Taq I (37 °C, 2 h) and Hae III (65 °C, 3 h) and then tested by 1.5% agarose gel electrophoresis. The products of 20JH1155 and *P. huashanica* obtained from S3-113 were cloned and sequenced to verify the source of homoeologous chromosomes. Sequence alignment by DNAMAN was performed on the data.

### 4.6. Variant Sequence Cloning and Analysis of 20JH1155

The published markers cannot amplify bands in our materials, so eleven pairs of primers were designed to clone the intron of TraesCS7A03G0411400 (*VRT-A2*) (IWGSC-RefSeq-v2.1). DNA was extracted from leaves at the trefoil stage by the Cetyltrimethyl Ammonium Bromide (CTAB) method. The PCR products were amplified in 1.5% agarose gel and purified by recovery kit (TIANGEN, Beijing, China) according to the directions. Then the products were connected to pMD^TM^19-T and converted to DH5α at heat shock. The sequences would be cultured on a solid LB medium and then picked single colonies to measure at AuGCT (Beijing AuGCT Biotechnology Co., Ltd., Beijing, China).

### 4.7. RNA Sequencing and Transcriptome Analysis

Immature spikes of three developmental points (Feekes 6,7,8) at the jointing stage and booting stage of 7182 and 20JH1155 were peeled and put in liquid nitrogen immediately for RNA extraction with three repetitions [21,58]. These samples were used to investigate gene expression about spikelet development (Appendix A. Transcriptome sequencing was performed using an Illumina NovaSeq 6000 platform at Beijing Biomarker Technologies Corporation (Beijing, China). The raw data were filtered and quality controlled to obtain high-quality clean data (FASTX-Toolkit), which further aligned to the common wheat reference genome (IWGSC RefSeq v2.1). The *de novo* transcriptome assembly was applied to the remaining unmapped reads (Trinity software v2.6.6). The normalized read counts were calculated fragments per kilobase of transcript sequence per million base pairs sequenced (FPKM) for each gene. Co-expression networks were constructed using the WGCNA package in R. The parameters were power 12, TOM-Type unsigned, min Module Size 30, deep Split 2, and merge Cut Height 0.25. Differentially expressed genes (DEGs) were calculated by DESeq2 (FPKM ≥ 1, Log_2_ |fold change| ≥ 1, adjusted *p* ≤ 0.05) [59]. Then the sequences make a comparison with *Th. ponticum*, *Th. elongatum*, *H. vulgare*, *S. cereale*, and durum wheat. Only unigenes found in *P. huashanica*, not wheat, and related species, were regarded as candidate genes. Gene ontology (GO) enrichment analysis was performed by cluster Profiler.

### 4.8. Molecular Marker Development Based on RNA-Seq

The unmapped reads of the samples were assembled to create a reference transcriptome for Ph#Ns using Trinity software (v2.8.4) and eliminated the 7182 unigenes with expression to get particular unigenes of 20JH1155. The top 500 unigenes were chosen according to the different expressions and compared with CS (IWGSC-RefSeq-v2.1). Sequences with 90% similarity to *P. huashanica* reference genome sequence would be compared with wheat-related materials, *Th. ponticum*, *Th. elongatum*, *Hordeum vulgare L.*, *S. cereale*, and T. *durum* to ensure the primers [60,61,62].

Primer sequences were synthesized at AuGCT (Beijing AuGCT Biotechnology Co. Ltd., Beijing, China). Then the markers were amplified in 20JH1155, common wheat 7182, CS, AK58, and wheat-related species, *Th*. *ponticum*, *Th. elongatum*, *H. vulgare*, *S. cereale*, and durum wheat to do PCR amplification. The products were detected on 8% polyacrylamide gel electrophoresis (PAGE).

## 5. Conclusions

In this study, we developed a stable elongated glume wheat-*P. huashanica* derived line 20JH1155 by cytogenetic analysis, GISH and sequential FISH, molecular marker, and transcriptome. 20JH1155, with additional two 3Ns chromosomes, has a short plant height, long spike length, a large grain weight, small and curly flag leaves, and resistance to FHB. The analysis of transcriptome data provides a perception of the gene expression of elongated glume. Based on RNA-seq, seven specific markers were developed to detect alien chromosomes of *P. huashanica.* In summary, our research broadens the genetic resource of wheat breeding and serves ideas for further study of long glume, large grains, and resistance to FHB.

## Figures and Tables

**Figure 1 ijms-24-06726-f001:**
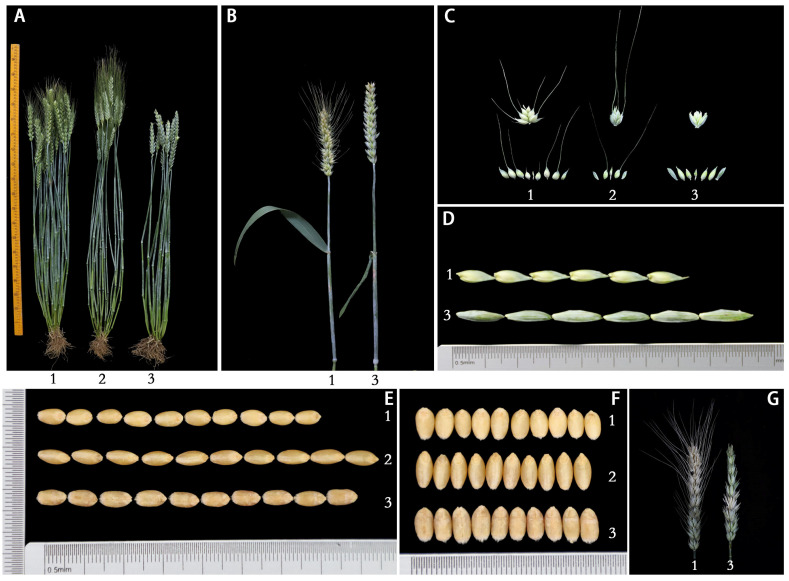
Agronomic traits of 20JH1155 and its wheat parents. The materials in the figures are (1) common wheat 7182; (2) *T. durum* Trs-372; (3) 20JH1155. (**A**) Adult plant. (**B**) Spike and peduncle. (**C**) Spikelet, glume, and lemma. (**D**) Glume. (**E**) 10-seed length. (**F**) 10-seed width. (**G**) FHB resistance to F. gPH1 of 7182 and 20JH1155.

**Figure 2 ijms-24-06726-f002:**
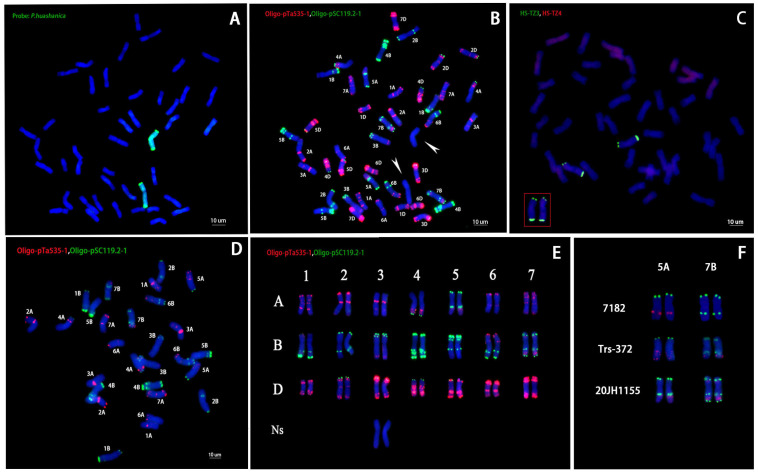
Cytological observation and in situ hybridization identification for chromosomes of root tip cell at mitotic metaphase of wheat-*P. huashanica* derivative 20JH1155. Chromosomes were dyed with counterstain DAPI (blue). (**A**) GISH analysis of 20JH1155 using *P. huashanica* genomic DNA as a probe (green) showed that there were 2 chromosomes of *P. huashanica*. (**B**) FISH karyotype using Oligo-pSc119.2 (green) and Oligo-pTa535 (red) of 20JH1155 showed that 20JH1155 contained 42 common wheat chromosomes and 2 *P. huashanica* chromosomes without signals (2 white arrows). (**C**) FISH karyotype using HS-TZ3 (green) and HS-TZ4 (red) of 20JH1155. The chromosomes in the red box have signals much like the 3Ns of *P. huashanica.* (**D**) FISH karyotype using Oligo-pSc119.2 (green) and Oligo-pTa535 (red) of Trs-372. (**E**) FISH karyotype of 20JH1155 referenced the published 7182 chromosomes. The number 1-7 meant 1-7 homoeologous, and A, B and D meant A genome, B genome and D genome of common wheat. Ns meant the genome of *P. huashanica.* (**F**) FISH signal comparison among 7182, Trs-372, and 20JH1155 in 5A and 7B chromosomes. There were green signals in 5AL and the end of 7BL of 20JH1155 which differed from its parental wheat 7182 and Trs-372.

**Figure 3 ijms-24-06726-f003:**
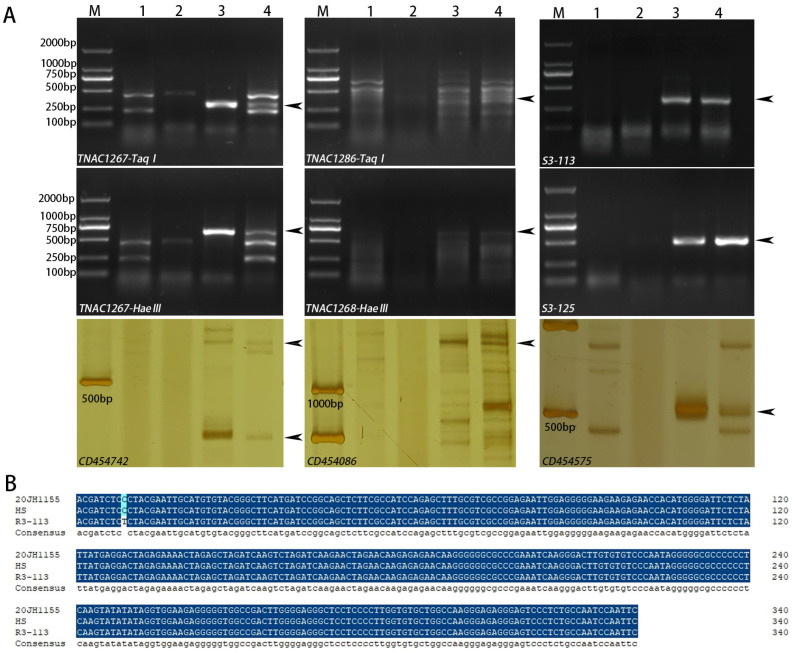
Molecular marker analysis of wheat-*P. huashanica* derived line 20JH1155. (**A**) EST, SSR, and PLUG markers, including TNAC 1267, TNAC1286, S3-113, S3-125, CD454742, CD454086, and CD454575, amplified the special bands of the third homoeologous. M: D2000. 1: 7182, 2: Trs-372. 3: *P. huashanica*. 4: 20JH1155. The black arrows indicate specific bands in the material and its alien parent. (**B**) Sequence alignment of 20JH1155, *P. huashanica*, and R3-113 by DNAMAN. HS means *P. huashanica,* and R3-113 is a special sequence of 3Ns in *P. huashanica*. The bases in dark blue indicated the same sequence of 20JH1155, *P. huashanica* and R3-113, and in light blue meant different bases.

**Figure 4 ijms-24-06726-f004:**
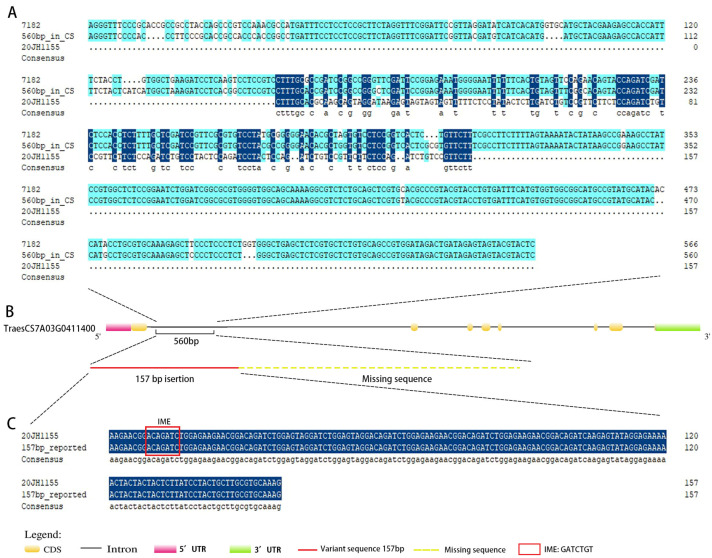
Sequence variation alignment of the cloning sequences of *VRT-A2*. The red box marked the motif, IME. (**A**) A 560-bp comparison among 7182, 20JH1155, and *TaVRT-A2*. Sequences in dark blue indicated the same bases of the three materials. The light blue showed the same bases of 7182 and Cs. (**B**) Sequence variation of *TaVRT-A2* gene between 7182 and 20JH1155. (**C**) Alignment of 20JH155 and the variant 157-bp reported in *T. polonicum*. The dark blue meant the same sequence of the two materials.

**Figure 5 ijms-24-06726-f005:**
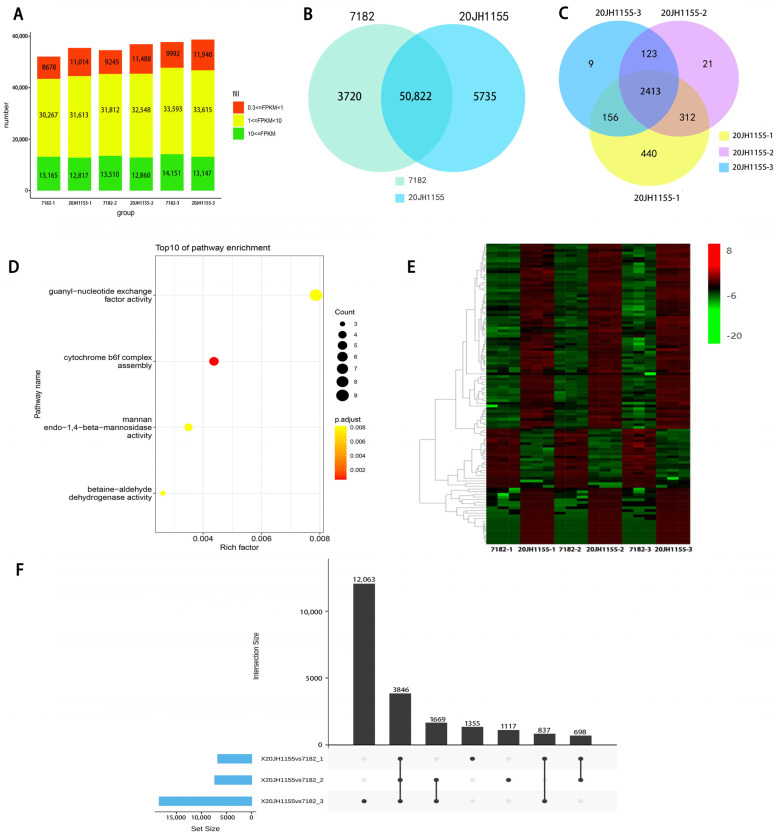
Transcriptome analysis of DEGs and analysis of 20JH1155 and 7182 in coregulate and expression. (**A**) Each sample has a number of genes with FPKM values of more than 0.3 from 52,110 to 58,702. (**B**) Venn-diagram of gene expression of 20JH1155 and 7182. (**C**) Venn diagram of potentially exogenous genes of 20JH1155 in three-time points. (**D**) GO enrichment analysis with potential alien genes expressing 2413 at all three time points. **(E)** The heatmap of annotated DEGs in the brown module of 18 samples. (**F**) An upset plot of DEGs overlapped in the three datasets.

**Figure 6 ijms-24-06726-f006:**
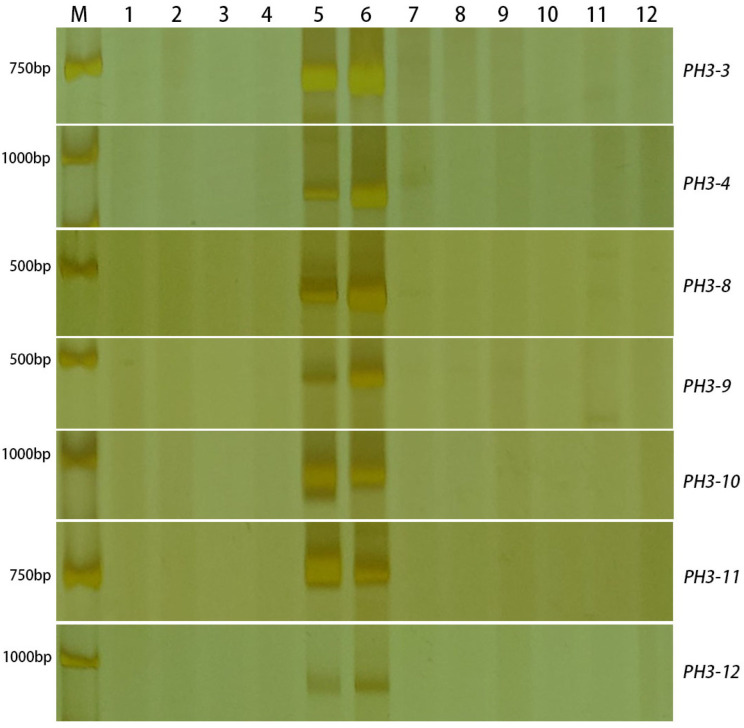
The amplification of the developed markers designed by RNA-seq DEGs. There are specific bands only on *P. huashanica* and 20JH1155 among common wheat and wheat-related species. M: D2000. 1: 7182. 2: CS. 3: AK58. 4: Trs-372. 5: *P. huashanica*. 6: 20JH1155. 7: *S. cerale*. 8: *Ae. geniculata.* 9: *Th. ponticum.* 10: *Th. elongatum.* 11: *Th. bessarabicum.* 12: *P. spicata*.

**Table 1 ijms-24-06726-t001:** Agronomic Traits of 7182, Trs-372, and 20JH11155.

Material	Plant Height (cm)	Spike Length (mm)	Glume length (mm)	Tiller Number	Spikelet Per Spike	TGW (g)	Grain Length (mm)	Grain Width (mm)	Awn Type
7182	81.8 ± 4.2 a	11.1 ± 1.1 ab	10.7 ± 0.9 b	15 ± 3 a	21 ± 1 a	36.63 ± 0.73 c	6.08 ± 0.03 c	3.19 ± 0.03 b	long
Trs-372	81.7 ± 2.5 a	10.5 ± 0.5 b	11.3 ± 0.5 b	12 ± 3 b	19 ± 1 b	53.13 ± 0.31 a	7.97 ± 0.02 a	3.31 ± 0.01 ab	long
20JH1155	70.5 ± 2.6 b	11.5 ± 4.9 a	15.7 ± 0.8 a	8 ± 2 c	19 ± 1 b	47.26 ± 0.38 b	7.25 ± 0.03 b	3.36 ± 0.02 a	zero

The letters a, b and c indicate signification differences among 7182, Trs-372 and 20JH1155 (*p* < 0.05).

## Data Availability

Not applicable.

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
