# Peer review of "Molecular Cytological Analysis and Specific Marker Development in Wheat-Psathyrostachys huashanica Keng 3Ns Additional Line with Elongated Glume"

_ijms, 2023, doi:10.3390/ijms24076726_

Round 1

Reviewer 1 Report

Wheat breeding using alien species as donors of useful agronomic traits, including resistance to various pathogens, is a promising direction, and the work of the authors is undoubtedly of great scientific and practical interest. The work is definitely relevant and undoubtedly important.
The article is multifaceted is extremely relevant and it is undoubtedly a high-level work and the authors obtained significant very interesting results. The article is very interesting, original, the content of the article corresponds to the abstract and title. The tables and figures are complementing the text well.

There are some comments and suggestions for authors.

1.    1. In the "Introduction" it is better to briefly formulate the aim of the work - one sentence - "the aim of our study was ...

2.     2. If F. graminearum strain PH-1 was used for resistance assays then you need to add a methodic to "materials and methods". In the section "materials and methods" there is no description of the analysis for resistance to Fusarium Head Blight.

3.     3. Please, check your "Supplementary Materials" for all tables and figures that you refer to in the text of the article. I didn't find Supplementary materials with all tables and figures.

4.     4. Figure 1-11 (Supplementary materials) - please, add designations M - molecular-weight size marker with size of fragments (bp), mark in the figure # or accessions name.

Other comments are presented in the text of the article

The authors have undoubtedly obtained important results, and I believe that the article can be published after revision.

Author Response

Dear Editors and Reviewers:

Thank you very much for your letter about your comments concerning our manuscript entitled “Molecular Cytological Analysis and Specific Marker Development of Wheat-Psathyrostachys huashanica Keng 3Ns Additional Line with Elongated glume” (ID: ijms-2296102). These comments are all valuable and very helpful for revising and improving our paper. We have studied comments carefully and made correction. Revised contents are marked in red in the text “revised-ijms-2296102”. The main corrections in the paper and the responds to the comments are as flowing:

Comment 1: In the "Introduction" it is better to briefly formulate the aim of the work - one sentence - "the aim of our study was ...

Response: Thank your suggestion and we have added this section as the comment at line 74-76. The additional sentence was “The aim of our study was to identify the superior traits and chromosomal composition of wheat-P. huashanica derived line 20JH1155, and explore the origin of the elongated glume trait.”

Comment 2: If F. graminearum strain PH-1 was used for resistance assays then you need to add a methodic to "materials and methods". In the section "materials and methods" there is no description of the analysis for resistance to Fusarium Head Blight.

Response: This advice helps a lot and we have added the description about assessment on fusarium head blight (FHB) at line 323-327: “The FHB resistance assessment was evaluated by F. graminearum strain PH1. At the flowering stage, single flower infusion method was used to infect 10 μL micro-conidial suspension (250,000 spores ml-1) to 10 spikes of 7182 and 20JH1155 respectively. Plastic bags were used to cover the spikes for 3 days, and the disease severity would be investigated 21 days later. The resistance of fusarium head blight (FHB) was estimated by infected spikelet rate (ISR).”

We have complemented the identification results at line 84-86: “Besides, the disease severity assessment of fusarium head blight (FHB) at 21d after infection was estimated by infected spikelet rate (ISR) showed that 20JH1155 (ISR=7.92%) had a higher resistance than 7182 (ISR=74.20) significantly (Figure 1G).”

Comment 3: Please, check your "Supplementary Materials" for all tables and figures that you refer to in the text of the article. I didn't find Supplementary materials with all tables and figures.

Response: Thank your suggestion about the Supplementary materials. We have summarized the tables into one before. And now we separated them into different tables including “Supplementary table 1, Supplementary table 2, Supplementary table 3, Supplementary table 4.”

Comment 4: Figure 1-11 (Supplementary materials) - please, add designations M - molecular-weight size marker with size of fragments (bp), mark in the figure # or accessions name.

Response: We are sorry for the negligence about forgetting to draw the M - molecular-weight size marker. All the marker is D2000, and we have added the marker in the figures and the accessions name. 

We have tried our best to improve the manuscript and made some changes in the manuscript. And we appreciate your warm work earnestly, and hope that the correction will meet with approval.

Once again, thank you very much for your comments and suggestions.

Sincerely yours,

Jingyu Pang

Reviewer 2 Report

This is a good manuscript on remote hybridization of wheat. However, it is written a little sloppy, with a number of typos and other details that need to be corrected. After correcting these details, the article can be published. In particular, please make the following corrections:

Line 150 … “partial homogenous” may have meant “partial homologous”?

Line 195 Ae. Geniculata - Ae. geniculata

Line 195 - 196 Th. besssrsbicum - Th. bessarabicum

Line 203  Ae. Geniculata - Ae. geniculata

Line 204 Th. besssrsbicum - Th. bessarabicum

Line 271 Ae. Geniculata - Ae. geniculata

Line 273 Th. besssrsbicum - Th. bessarabicum

Line 274 Pseudoroegeria spicata - Pseudoroegneria spicata

Line 349 Moecular - Molecular

For the abbriviations, it is necessary to spell out the full term at its first mention, for instance for the abbreviation FHB.

Regarding FHB data, the description of the FHB protocol and results should be extended. The Methods section should be supplemented with an FHB resistance analysis protocol. How exactly was resistance assessed? What were the results? There is only one picture in the article, and it is not self-explaining, thus the description should be placed in the text.

Author Response

Dear Editors and Reviewers:

Thank you very much for your letter about your comments concerning our manuscript entitled “Molecular Cytological Analysis and Specific Marker Development of Wheat-Psathyrostachys huashanica Keng 3Ns Additional Line with Elongated glume” (ID: ijms-2296102). These comments are all valuable and very helpful for revising and improving our paper. We have studied comments carefully and have made correction. Revised contents are marked in red in the paper. The main corrections in the paper and the responds to the comments are as flowing:

Comment 1:

(1) Line 150 … “partial homogenous” may have meant “partial homologous”?

Response: The line 150 word “homogenous” was modified to line 166 word “homologous”.

(2) Line 195 Ae. Geniculata - Ae. geniculata

Response: The line 195 word “Ae. Geniculata” was edited to line 221 word “Ae. geniculata”.

(3) Line 195 - 196 Th. besssrsbicum - Th. bessarabicum

Response: The line 195 word “Th. besssrsbicum” was altered to line 221-222 “bessarabicum”.

(4) Line 203 Ae. Geniculata - Ae. geniculata

Response: The line 203 word “Ae. Geniculata” was edited to line 229 “Ae. geniculata”.

(5) Line 204 Th. besssrsbicum - Th. bessarabicum

Response: The line 204 word “Th. besssrsbicum” was altered to line 230 “Th. bessarabicum”.

(6) Line 271 Ae. Geniculata - Ae. geniculata

Response: The line 271 word “Ae. Geniculata” was edited to line 307 “Aegilops geniculata”.

(7) Line 273 Th. besssrsbicum - Th. bessarabicum

Response: The line 273 word “Th. besssrsbicum” was edited to line 309 “Th. bessarabicum”.

(8) Line 274 Pseudoroegeria spicata - Pseudoroegneria spicata

Response: The line 274 word “Pseudoroegeria spicata” was changed to line 309 word “Pseudoroegneria spicata”.

(9) Line 349 Moecular - Molecular

Response: The line 349 word “Moecular” was edited to line 392 “Molecular”.

Comment 2: For the abbriviations, it is necessary to spell out the full term at its first mention, for instance for the abbreviation FHB.

Response: We are sorry for the negligence about forgetting full name at the first mention of abbriviations. We have added the full name of the abbreviation FHB at line 85: “Besides, the disease severity assessment of fusarium head blight (FHB) at 21d after infection was estimated by infected spikelet rate (ISR) showed that 20JH1155 (ISR=7.92%) has a higher resistance than 7182(ISR=74.20) significantly (Figure 1G).”

Comment 3: Regarding FHB data, the description of the FHB protocol and results should be extended. The Methods section should be supplemented with an FHB resistance analysis protocol. How exactly was resistance assessed? What were the results? There is only one picture in the article, and it is not self-explaining, thus the description should be placed in the text.

Response: This advice helps a lot and we have added the description about assessment on fusarium head blight (FHB) at line 323-327: “The FHB resistance assessment was evaluated by F. graminearum strain PH1. At the flowering stage, single flower infusion method was used to infect 10 μL micro-conidial suspension (250,000 spores ml-1) to 10 spikes of 7182 and 20JH1155 respectively. Plastic bags were used to cover the spikes for three days, and the disease severity would be investigated 21 days later. The resistance of fusarium head blight (FHB) was estimated by infected spikelet rate (ISR).”

We have complemented the identification results at line 84-87: “Besides, the disease severity assessment of fusarium head blight (FHB) at 21d after infection was estimated by infected spikelet rate (ISR) showed that 20JH1155 (ISR=7.92%) has a higher resistance than 7182(ISR=74.20) significantly (Figure 1G).”

Other changes marked in red are presented in the text “revised-ijms-2296102”.

We would like to express our great appreciation to you for comments on our paper. Looking forward to hearing from you. Thank you and best regards.

Sincerely yours,
